

# A retrospective study of the correlation between high serum ferritin levels and the risk of gestational diabetes mellitus in midpregnant women

Xizhenzi Fan[1,*], Pan Zhang[2,*], Lingli Wang[1], Wenhui Song[1], Achou Su[1] and Tianxiao Yu[1]

[1] Research Center for Clinical Medical Sciences, The Fourth Hospital of Shijiazhuang, Shijiazhuang, China
[2] Department of Clinical Laboratory, The Fourth Hospital of Shijiazhuang, Shijiazhuang, China
[*] These authors contributed equally to this work.

Corresponding author
Tianxiao Yu,
yutianxiao1111@163.com

## ABSTRACT

**Aims.** Gestational diabetes mellitus (GDM) is any degree of glucose intolerance with onset or first detection during pregnancy, with an inconsistent association with serum ferritin (SF). We aimed to ascertain the relationship between SF and the risk of GDM in mid-pregnancy and provide evidence for implementing clinical individualized and reasonable iron supplementation regimens.

**Methods.** A retrospective study was conducted to investigate 1,052 pregnant women at 24–28 weeks of gestation who were examined in the obstetrics department of The Fourth Hospital of Shijiazhuang from January 2019 to December 2021. Questionnaires were used to obtain the general information. The levels of serum ferritin (SF), serum calcium, glycated haemoglobin (HbA1c) and Oral-Glucose-Tolerance-Test (OGTT) were reviewed. The GDM was diagnosed by glucose tests. Multivariate logistic regression was used to determine the relationship between serum ferritin and GDM.

**Results.** Compared to the non-GDM group, the GDM group had significantly higher level of SF (13.95 (8.59–23.65) ng/mL *vs.* 12.11 (7.27–19.86) ng/mL, ($p = 0.012$)). While SF levels positively correlated with 1-hour plasma glucose levels (PG1H) ($r = 0.061$, $p = 0.047$), there was a negative correlation between SF and HbA1c levels ($r = -0.078$, $p = 0.011$). The risk of GDM with higher levels of SF was increased (1.010 (95% CI [1.001–1.020], $p = 0.025$)). In the univariate logistic regression model, the risk of GDM in pregnant women with high ferritin levels was 1.010 (95% CI [1.001–1.020], $p = 0.025$). After adjustment for age and pre-pregnancy body mass index (BMI), the risk of GDM was significantly increased by 44% and 42% respectively (adjusted odds ratio (AOR) = 1.440, 95% CI [1.025–2.023], $p = 0.035$), AOR = 1.420 (95% CI [1.011–1.995], $p = 0.043$). After multivariate adjustment for age and pre-pregnancy BMI, the results were moderately revised (AOR = 1.427, 95%CI [1.013–2.008], $p = 0.042$).

**Conclusions.** Elevated SF levels of mid-pregnancy was associated with risk of GDM, which may guide the implementation of pregnancy-specific supplementation to some extent with the support of further clinical trials.

## INTRODUCTION

Gestational diabetes mellitus (GDM) is defined as any degree of glucose intolerance with onset or first recognition during pregnancy (*Plows et al., 2018*). Previous studies have shown that GDM not only increases the risk of pregnant women who have type 2 diabetes (T2DM) and cardiovascular disease (CVD) but can also lead to a variety of adverse maternal and neonatal outcomes, such as pre-eclampsia, pregnancy infection, acute complications of diabetes and neonatal jaundice, *etc.* (*Zaugg et al., 2020*; *Li et al., 2021*). The mechanisms involved in the development of GDM are multifactorial and have not been fully understood. It is widely accepted that age, race and family history of diabetes are known risk factors for GDM.

Recently, some studies suggest that some vitamins and trace elements may also play an important role in the development of GDM (*Wang et al., 2022*; *Eroglu et al., 2021*; *Osorio-Yanez et al., 2017*). For example, iron, as one of metals essential for cellular functions, plays an important role in ensuring the newborn's maturity and avoiding adverse pregnancy outcomes (*Cheng et al., 2020*; *Low et al., 2016*). However, as an oxidizing active metal trace element, iron can also catalyze several cellular reactions, producing reactive oxygen species, which can contribute to insulin resistance, followed by the reduction of insulin secretion and the induction of GDM (*Means, 2020*).

Serum ferritin (SF), the principal iron storage protein, provides an indicator that directly reflects the state of iron in the body and plays a crucial role in iron metabolism. It is widely used as a routinely available indicator to assess iron status (*Daru et al., 2017*). However, the cut-off value of SF in pregnancy with iron deficiency has not been explicitly established, and there are conflicting opinions on whether excessive SF is associated with GDM (*Rawal et al., 2017*; *Khambalia et al., 2016*; *Helin et al., 2012*; *Bowers et al., 2011*).

The study aimed to explore the relationship between SF levels and the risk of GDM in mid-gestation with a large sample size, thereby providing epidemiological evidence for the development of individualized iron supplementation strategies to prevent over-supplementation-induced adverse pregnancy outcomes in GDM.

## MATERIALS AND METHOD

### Study participants

This retrospective study was obtained from the clinical data of pregnant women who had visited the Fourth Hospital of Shijiazhuang City for obstetric examination from January 2019 to December 2021. All participants were informed and signed informed consent when enrolled. The Ethics Review Committee approved the study of The Fourth Hospital of Shijiazhuang (20220029). Eligible participants were women with a single pregnancy; age 18–45; mid-trimester (24–28 weeks), ferritin screening, 75 g Oral-Glucose-Tolerance-Test. Pregnant women with chronic diseases such as type 1 or type 2 diabetes, hypertension, malignancy, hypothyroidism and acute or chronic inflammatory or infectious diseases such as hepatitis B were excluded from this study (*Yan et al., 2020*). To reduce the interferences of other factors such as inflammation on high SF levels, blood tests such as white blood cells (WBC), neutrophils (NEUT), lymphocytes (LYMPH) and eosinophils (EO) were

normal for all pregnant women enrolled. Finally, 1,052 pregnant women were enrolled in this study, of which 197 were GDM and 855 were normal glucose controls.

## Diagnosis of GDM

According to the World Health Organization (WHO) guideline (*Anonymous, 2014*), GDM is diagnosed when fasting plasma glucose (FPG) is 5.1–6.9 mmol/L or 1-hour plasma glucose (PG1H) $\geq$10.0 mmol/L or 2-hour plasma glucose (PG2H) is 8.5–11 mmol/L at 24–28 weeks of gestation. Based on the above criteria, participants were divided into two groups, 197 cases with GDM and 855 pregnant women with normal glucose plasma levels.

## Measurements

Obstetricians measured the height of the pregnant women and used a questionnaire to record characteristics such as their age, pre-pregnancy weight, annual household income, adverse pregnancy history, exposure to second-hand smoke, and alcohol consumption during pregnancy from all pregnant women.

Blood samples were collected during a routine obstetric examination, which refrigerated and centrifuged at 4 °C by a laboratory physician. Plasma glucose was measured at fasting and 1 h and 2 h after the 75 g glucose load. An automatic biochemical analyzer (Cobas 701; Roche) was used to measure the plasma glucose and serum calcium, and serum ferritin was measured with an Abbott i2000 automated immune analyzer. HbA1c was measured using the automatic glycosylated haemoglobin analyzer (HA-8180).

## Statistical analysis

Statistical analysis was performed with SPSS 22.0 software (IBM SPSS Statistics, version 22). The distribution of baseline data for continuous variables is expressed as the median (interquartile range), and categorical variables were presented by frequencies (percentage). The Mann–Whitney U test was used for non-parametric variables and the chi-square test for categorical variables. Spearman correlation analysis was used to analyze the correlation between ferritin and glucose metabolism indexes. Meanwhile, multivariate logistic regression were used to determine the effect of high serum ferritin on the risk of GDM. The crude/adjusted odds ratios (OR/AOR) and 95% confidence intervals (95% CI) from the logistic regression coefficients and corresponding covariance matrices were computed to analyze the relationship between serum ferritin level and GDM. $P < 0.05$ was considered statistically significant.

## RESULTS

In this study of 1,070 subjects, 1,052 subjects underwent Oral-Glucose-Tolerance-Test at 24–28 weeks of pregnancy, and 18 women did not follow our study (according to the inclusion and exclusion criteria). The participants were divided into two groups according to their blood glucose concentrations of OGTT. Figure 1 shows the flow chart of this study. Table 1 describes the basic characteristics of pregnant women which may be associated with GDM. Women who developed GDM during pregnancy were older (30 (28–33) *vs.* 29 (26–31), $p < 0.0001$), and they had higher pre-pregnancy weight (58.5 (54–63)

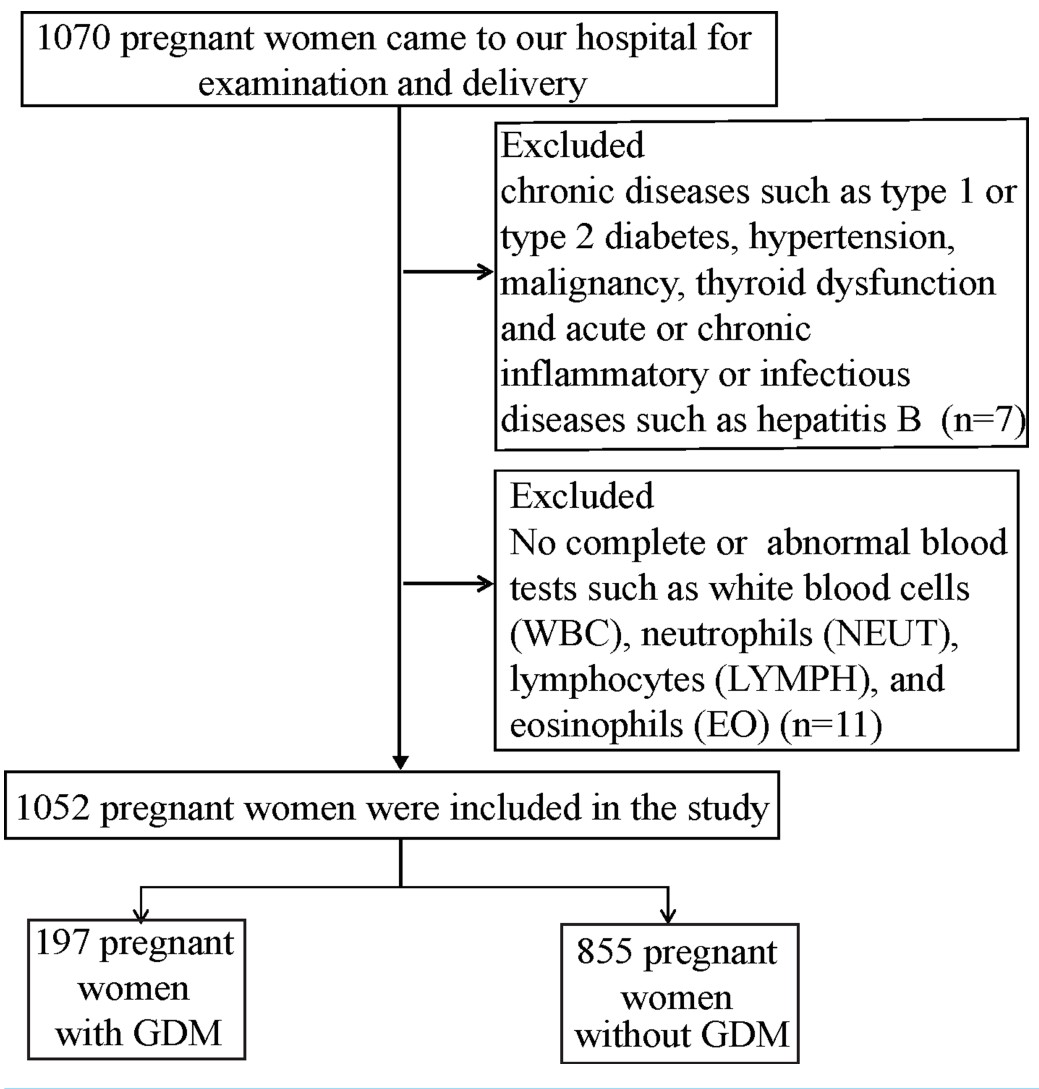

**Figure 1**  Flow chart of the study population.

*vs.* 56.0 (51–62) kg, $p = 0.002$) and higher pre-pregnancy body mass index (BMI) (22.4 (20.7–24.5) *vs.* 21.2 (19.5–23.5) kg/m$^2$, $p < 0.0001$) than women without GDM. There was no significant difference in height, education, annual family income, number of births, adverse pregnancy history, pre-pregnancy drinking and second-hand smoke exposure ($p > 0.05$).

The results of blood tests at 24–28 gestational weeks were presented in Table 2. Overall, there was no difference in serum calcium concentration between the two groups ($p = 0.888$), whereas the differences were statistically significant for the levels of FPG (5.1 (4.8–5.3) mmol/L *vs.* 4.6 (4.4–4.8) mmol/L, ($p < 0.0001$)), PG1H (9.1 (7.8–10.1) mmol/L *vs.* 7.2 (6.3–8.0) mmol/L, ($p < 0.0001$)) and PG2H (8.0 (6.9–8.8) mmol/L *vs.* 6.3 (5.8–6.9) mmol/L, ($p < 0.0001$)). HbA1c was slightly increased in pregnant women with GDM (5.3 (5.2–5.6) % *vs.* 5.3 (5.1–5.5) %, ($p = 0.004$)), although the change was significant.

**Table 1   Basic characteristics of pregnant women ($n = 1,052$).**

| Characteristics | GDM | NON GDM | *p*-value |
|---|---|---|---|
| n | 197 | 855 | |
| Age (years) | 30 (28–33) | 29 (26–31) | 0.000 |
| Height (cm) | 162 (160–165) | 162 (160–165) | 0.661 |
| Pre weight (kg) | 58.5 (54–63) | 56.0 (51–62) | 0.002 |
| Pre pregnancy BMI (kg/m$^2$) | 22.4 (20.7–24.5) | 21.2 (19.5–23.5) | 0.000 |
| Years of education (n%) | | | 0.447 |
|    <16 years | 105 (19.6) | 430 (80.4) | |
|    ≥16 years | 92 (17.8) | 425 (82.2) | |
| Family yearly income (n%) | | | 0.835 |
|    <100,000 CNY | 103 (19.0) | 440 (81.0) | |
|    ≥100,000 CNY | 94 (18.5) | 514 (81.5) | |
| Nulliparous (n%) | 133 (18.5) | 586 (81.5) | 0.340[a] |
| Adverse pregnancy history (n%) | 55 18.8) | 237 (81.2) | 0.955 |
| Drinking before pregnancy (n%) | 6 (26.1) | 17 (73.9) | 0.415[a] |
| Second-hand smoking (n%) | 29 (25.4) | 85 (74.6) | 0.052 |

Notes.
   Data of continuous variables were expressed as median (interquartile range) and measured by Mann–Whitney U.
   Categorical variables were presented as frequency (%) and measured by Chi-square test. BMI, body mass index; CNY,
   Chinese yuan; GDM, gestational diabetes mellitus.
   [a]Based on Fisher's exact test.

**Table 2   Comparison of biochemical indices between women with GDM and NON GDM based on their OGTT results.**

| Variables | GDM | NON GDM | *p*-value |
|---|---|---|---|
| FPG (mmol/L) | 5.1 (4.8–5.3) | 4.6 (4.4–4.8) | <0.000 |
| PG1H (mmol/L) | 9.1 (7.8–10.1) | 7.2 (6.3–8.0) | <0.000 |
| PG2H (mmol/L) | 8.0 (6.9–8.8) | 6.3 (5.8–6.9) | <0.000 |
| HbA1c (%) | 5.3 (5.2–5.6) | 5.3 (5.1–5.5) | 0.004 |
| Serum ferritin (ng/mL) | 13.95 (8.59–23.65) | 12.11 (7.27–19.86) | 0.012 |
| Serum Ca (mmol/L) | 2.15 (2.08–2.21) | 2.15 (2.08–2.21) | 0.888 |

Notes.
   OGTT, oral glucose tolerance test; FPG, fasting plasma glucose; PG1H, plasma glucose 1 h after the oral glucose tolerance
   test; PG2H, plasma glucose 2 h after the oral glucose tolerance test; HbA1c, glycated hemoglobin/hemoglobin A1c.
   Values were shown as median (interquartile range) for continuous variables and measured by Mann–Whitney U test.

Our findings indicate that, the SF level was found to be higher in pregnant women with GDM, and the difference was statistically significant (13.95 (8.59–23.65) ng/mL *vs.* 12.11 (7.27–19.86) ng/mL, ($p = 0.012$)).

In Table 3, we examined the linear relationships between SF and serum glucose levels at different time intervals in the OGTT test, including FPG, PG1H, PG2H, and HbA1c in pregnant women. The results of the Spearman correlation analysis indicated a positive correlation between SF and PG1H ($r = 0.061$, $p = 0.047$), while a negative correlation was observed between SF and HbA1c ($r = -0.07$). However, no correlation was observed between SF and FPG ($r = 0.026$, $p = 0.396$) or PG2H ($r = 0.056$, $p = 0.072$).

**Table 3 Correlation analysis between serum ferritin (SF) and glucose metabolism index in pregnant women.**

| Variables | FPG | | PG1H | | PG2H | | HbA1c | |
|---|---|---|---|---|---|---|---|---|
| | r | p | r | p | r | p | r | p |
| Serum ferritin (SF) | 0.026 | 0.396 | 0.061 | 0.047 | 0.056 | 0.072 | −0.078 | 0.011 |

Notes.

The Spearman correlation analysis were performed to assess the correlation between serum ferritin and glucose metabolism index in pregnant women.

**Table 4 Univariate and multiple regressions between serum ferritin (SF) and GDM (odds ratios and 95% confidence intervals).**

| Dependent Variable | Independent Variable | OR | 95% CI | p |
|---|---|---|---|---|
| GDM | Ferritin, ≤20 reference, >20 | 1.010 | 1.001–1.020 | 0.025[a] |
| GDM | Ferritin, Age, <30 reference, >30 | 1.440 | 1.025–2.023 | 0.035[b] |
| GDM | Ferritin, BMI, <24 reference, >24 | 1.420 | 1.011–1.995 | 0.043[c] |
| GDM | Ferritin, Age, BMI | 1.427 | 1.013–2.008 | 0.042[d] |

Notes.

[a] Ferritin concentration was calculated to be the 75th percentile for healthy pregnant women. Univariate logistic regression model result between ferritin ≤ 20 ng/mL group and ferritin >20 ng/mL group.
[b] Multivariate logistic regression model result adjusted for age.
[c] Multivariate logistic regression model result adjusted for pre-pregnancy BMI.
[d] Multivariate logistic regression model result adjusted for age and pre-pregnancy BMI.

Furthermore, we performed the logistic regression analyses to determine the influence of elevated serum ferritin on the risk of GDM, the results are shown in Table 4. Considering the ferritin concentration of the 75th percentile for healthy pregnant women as the cut-off point to define high ferritin, the logistic regression results suggested that ferritin concentrations were significantly associated with the risk of GDM. In the univariate logistic regression model, the risk of GDM in pregnant women with ferritin levels above 20 ng/ml was 1.010 (95% CI [1.001–1.020], $p = 0.025$). After adjusting for age, the risk of GDM was significantly increased by 44% (adjusted odds ratio (AOR) = 1.440, 95% CI = [1.025–2.023], $p = 0.035$), and after adjustment with pre-pregnancy BMI, AOR was 1.420 (95% CI [1.011–1.995], $p = 0.043$). The results were moderately modified after multivariable adjustment with age and pre-pregnancy BMI, and the AOR was 1.427 (95% CI [1.013–2.008], $p = 0.042$). Based on the above results, our study indicated that the risk of GDM increased by 42.7% in pregnant women whose ferritin concentration exceeded 20 ng/ml in mid-pregnancy.

## DISCUSSION AND CONCLUSION

Iron has a biological function in haemoglobin formation, oxidative stress and immune response and is one of the essential trace elements in maintaining the human body's normal physiological processes (*Georgieff, 2020*). Due to the increase in placental growth and fetal nutrition during pregnancy (*Cao & Fleming, 2016*), iron deficiency has become a common global public health problem related to adverse pregnancy outcomes. Serum ferritin has 90 percent sensitivity and 85 percent specificity in estimating body iron stores and is used as an indirect screening tool for iron deficiency (*Torti & Torti, 2002*). Recently, epidemiological

evidence demonstrates that SF levels are not only the biomarker of iron storage in the body, it is also associated with many chronic inflammation-related diseases, such as cardiovascular diseases (CVDs) and diabetes (*Durrani et al., 2021*). Studies have shown that the ferritin levels are associated with abnormal glucose tolerance metabolism, GDM or the development of T2DM in the distant future (*Kunutsor et al., 2013*). Some previous studies had relatively small sample sizes (*Zein et al., 2015*) (less than 700 participants), and only a few cases of GDM (3.3% incidence) were reported (*Chen, Scholl & Stein, 2006*). We performed a large retrospective study to obtain data that would provide stronger scientific evidence for the Chinese population. Compared with the subjects of other studies, this study had higher SF levels in GDM groups, which aligned with results of *Zhang et al. (2021)*. Our study obtained results consistent with most previous studies that plasma ferritin concentration was positively associated with GDM, such as *Krisai et al. (2016)* and *Zein et al. (2015)*, which showed a positive relationship between ferritin and PG1H. We found an interesting result that at gestational 24–28 weeks, HbA1c levels, representing blood glucose levels over the past 2–3 months, were significantly negatively correlated with SF levels, which is consistent with *Hashimoto et al. (2008)* study. Similar conclusions were observed by *Shanthi et al. (2013)*, *Coban, Ozdogan & Timuragaoglu (2004)*, and *Cheng et al. (2020)* in a study of patients with iron deficiency anemia without a history of diabetes. We believe that since HbA1c represents the serum glucose levels of the past 2–3 months, which are strongly influenced by various other coexisting factors, and *Phelps et al. (1983)* have shown biphasic changes in HbA1c levels during pregnancy, with a nadir at 24 weeks' gestation. Therefore, the exact relationship between HbA1c and SF needs to be further discussed in the absence of confounding factors. However, our large sample study found that SF levels were positively correlated with PG1H and negatively correlated with HBA1c. The present study suggests that ferritin may be involved in the occurrence and development of GDM. This study analyzed correlations between serum ferritin and several known risk factors of GDM by univariate and multiple logistic regressions. As demonstrated in many studies, older and overweight women had significantly increased risks for GDM (*Lende & Rijhsinghani, 2020*). There was a significant difference in our studied groups in terms of age and pre-pregnancy BMI. There was a positive association between serum ferritin levels and the risk of GDM despite adjustment for age and pre-pregnancy BMI. Consequently, our large sample study revealed that elevated serum ferritin is an independent risk factor for GDM in mid-pregnancy.

Our study has some limitations. Diabetes is strongly associated with systemic inflammatory states, and it is well known that higher CRP (C reactive protein) may increase the risk of developing diabetes (*Rohm et al., 2022*). Pregnancy is a chronic inflammatory process, and ferritin will also increase significantly in the state of chronic inflammation (*Davies et al., 2021*). Due to the limitations of the retrospective study, the results of CRP, which can reflect the inflammatory state in the body, could not be obtained for all pregnant women, especially for control pregnant women with normal blood glucose levels. We can only make the results more scientifically meaningful by strictly standardizing the population inclusion criteria based on the screening principles of other similar studies to reduce the impact of confounding factors on the reliability of the results. In addition, because the study

measured SF and GDM at the same time, the causal relationship between SF and GDM could not be clearly established, which was also one of the limitations of our study. And we did not include women diagnosed with GDM after 28 weeks of gestation. Therefore, although our conclusions suggest that an elevated ferritin level in mid-pregnancy is an independent risk factor for GDM, there is a need for further prospective cohort studies to consistently observe the association of SF with GDM throughout pregnancy and to include baseline ferritin status and iron intake during pregnancy for further analysis.

In conclusion, the ferritin level can play a relatively scientific predictive position in the development of GDM and information the implementation of pregnancy-specific supplementation regimens, though the current conclusion need to be supplemented by evidence of causation.

### Funding

The study was funded by the Key Medical Scientific Research Project of Hebei province (No. 20231652). The funders had no role in study design, data collection and analysis, decision to publish, or preparation of the manuscript.

### Grant Disclosures

The following grant information was disclosed by the authors:
Key Medical Scientific Research Project of Hebei province: No. 20231652.

### Competing Interests

The authors declare there are no competing interests.

### Author Contributions

- Xizhenzi Fan conceived and designed the experiments, performed the experiments, analyzed the data, prepared figures and/or tables, authored or reviewed drafts of the article, and approved the final draft.
- Pan Zhang conceived and designed the experiments, performed the experiments, analyzed the data, prepared figures and/or tables, and approved the final draft.
- Lingli Wang performed the experiments, analyzed the data, prepared figures and/or tables, and approved the final draft.
- Wenhui Song analyzed the data, prepared figures and/or tables, and approved the final draft.
- Achou Su analyzed the data, prepared figures and/or tables, and approved the final draft.
- Tianxiao Yu conceived and designed the experiments, authored or reviewed drafts of the article, and approved the final draft.

### Human Ethics

The following information was supplied relating to ethical approvals (i.e., approving body and any reference numbers):

The Ethics Review Committee approved the study of The Fourth Hospital of Shijiazhuang (20220029).
## Data Availability

The raw data are available in the Supplemental File.

## Supplemental Information

Supplemental information for this article can be found online at http://dx.doi.org/10.7717/peerj.18965#supplemental-information.

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
