# Peer review of "A retrospective study of the correlation between high serum ferritin levels and the risk of gestational diabetes mellitus in midpregnant women"

_PeerJ, doi:10.7717/peerj.18965_

## Round 0.1 · original submission · Major Revisions

· Academic Editor

Major Revisions

Although the topic of this study is important, there are several limitations in study design, statistical analysis and data reporting, as pointed out by the Reviewers. The authors omitted some important blood tests. For instance, they analyzed serum calcium but not iron; they analyzed serum HbA1c but not hemoglobin.

Proper statistical tests were not used. For instance, the Spearman test was used instead of Pearson test for parametric variables. The data are not properly reported. For instance, the standard deviation is not given with the mean values (only range is given). In Table 2, HbA1c levels are 5.3% for both the groups, yet their P value shows that they are significantly different. In the same table, serum calcium levels in both the groups are identical so it is very less likely that the data are not normally distributed. In this case, a parametric test should have been used or some evidence be given to prove non-normal distribution.
The authors concluded that serum ferritin level can play a relatively scientific predictive position in the development of GDM. In fact, some previous studies argued that diabetes mellitus causes increased serum ferritin concentrations rather than increased serum ferritin concentrations cause diabetes. It has also been commented earlier that serum ferritin is glycosylated and this glycosylation affects the half-life of this protein and, therefore, its serum concentrations increase.

The authors have previously published quite a similar study. They should clearly point out as how this work is different from their previous publication to justify the rationale of this study.

·

Basic reporting

No

Experimental design

Abstract:
Line 22: What was the study design?
Line 22-23: What was the gestational age of the participants at the time of recruitment? As the OGTT is done in the second trimester of pregnancy, how did you assess the risk of GDM in mid-pregnancy? Did you take the data in 1st trimester of pregnancy? Please clarify the point.
Line 26: Were the participants healthy pregnant women?
Line 26: Were the control group healthy pregnant women?

Validity of the findings

No

Additional comments

None

Reviewer 2 ·

Basic reporting

Please see report

Experimental design

Please see report

Validity of the findings

Please see report

Additional comments

Please see report

Annotated reviews are not available for download in order to protect the identity of reviewers who chose to remain anonymous.

·

Basic reporting

Abstract:
Line 34: The sentence “the results were moderately modified” is unclear. Please edit the verb.
In general, please edit the grammatical errors in the entire manuscript and tables.

Study participants:
Line 96: It would be better if you explain the definition of GDM and its criteria before your biochemical analysis subsection.

Results:
Lines 129-135: please delete excessive parentheses.
Line 170: Please edit “concurred with Zhang” into “aligned with results of Zhang et al.” or “aligned with Zhang et al. findings”. Please add “et al.” to all your citations and keep the same format (full names or name abbreviations).
Lines 172-173: Edit “Philipp Krisaiand” and “Zein S”, (add et al.) and please remain consistent when you cite other studies (E.g. name abbreviations or full names).
Lines 174, and 177: please edit citations.
Line 177: Edit “and his co-workers” into “et al.” or “and his colleagues”


Statistical analysis:
Lines 104-105: Please edit “median (quartile1- quartile3)” into “median (Inter-quartile range)”. And in the whole manuscript.
Lines 106-107: Edit grammatical errors. And change “non-normal” into “Nonparametric”.
Line 117: Edit OGTT.

Table 3: Please edit the grammatical errors.

Experimental design

Abstract:
Line 17: Please describe GDM definition in first line and also your GDM criteria in the abstract in one or 2 lines (e.g. glucose tests, etc.).

Study participants:
Lines 78-79: Please explain why you chose mid-pregnancy, and did you include other GDM subjects after 28 weeks of gestation? GDM can last till the end of pregnancy.
Line 79: Please specify the type of OGTT test (e.g. 2 hours, 120 min etc.) and also edit the abbreviations (e.g. OGGT – 120 min) and it the entire manuscript and tables.
Lines 82-85: Did you exclude the subjects who had elevated WBCs? And also, please explain if you have done the same thing for your control group and define your control group as well (e.g. age, FBS or OGTT test, etc.). have you excluded subjects with other conditions (for example: previous GDM, multi-pregnancy, hypothyroidism, etc.)? Please define your inclusion and exclusion criteria more precisely.

Data collection:
Line 87: Please specify if you have used a specific questionnaire or you have designed your own questionnaire.

Statistical analysis:
Line 118: The sentence is unclear. Did you exclude them from the initial sample size or lost follow-up (as it was a retrospective study). And again, the inclusion and exclusion criteria are not clearly defined.
Lines 119-120: Please edit “blood glucose concentrations of OGTT”

Validity of the findings

Introduction:
Lines 60-61: This sentence is pre-judgmental “To date, we have not identified the appropriate serum iron concentration…” It is not completely agreed that Iron or excess Iron would have an adverse effect on GDM.

Diagnosis of GDM:
Line 100: How you define “matched”? according to your results, two groups are not matched for age/weight/BMI!!
So please define “Matched groups”.

Statistical analysis:
Line 120: the sentence “Table1 describes the basic characteristics of pregnant women which may be associated with GDM” is more like a guess. You haven’t reported your results, so it is pre-judgmental.
Table1 and table 2: Please specify which tests are Mann-Whitney and which are parametrical.
Table 4: Please double-check the “greater than or <” and “less than or >” signs in Table 4 and make sure they are correct.

Results:
Line 133: please remove “to our interest” why you are interested? Researchers should be unbiased. Scientific articles aim to communicate research findings in a clear and unambiguous manner. For example, you can use: our findings indicate that…
Lines 157-159: “Due to the increase in placental growth and fetal nutrition…” please provide a reference for this.
Line 165: Please change “Currently” into that particular study that concluded that result (Reference 20).
Lines 185-186: Differences in age and baseline BMI in your groups are not considered findings. Please edit the sentence to avoid misleading (e.g. there was a significant difference in our studied groups in terms of age and BMI)
Lines 196-197: The meaning of this sentence is unclear “so it 196 was difficult to obtain complete data, which unfortunate that GDM causes high ferritin levels, or 197 higher ferritin levels cause GDM cannot be further explained” please edit the sentence.

Additional comments

This is a novel subject as there is not much data regarding the impact of Iron and Iron supplementation on gestational diabetes. The authors are not native English speakers and some of the errors are because of language barriers and the rest is due to lack of experience in scientific writings. I cannot comment on statistical results as theirs is no definitive inclusion and exclusion criteria, and I am not a statistical expert. I can check the statistical methods and see if the results are correct, but it is better to ask from a statistical expert as well. But the results are satisfactory after editing the manuscript.

Reviewer 4 ·

Basic reporting

The manuscript explores the relationship between serum ferritin (SF) levels and the risk of gestational diabetes mellitus (GDM) in mid-pregnancy, which addresses an important public health concern. The study's large sample size and focus on a significant clinical outcome, GDM, make this research valuable. However, several areas require attention before it can be recommended for publication. Below are my detailed comments:

Line 132: You’ve already provided the full name of HbA1c earlier, so there is no need to repeat it here. Please use the abbreviation consistently throughout the manuscript.

Table 2: If the p-value is less than 0.001, it should be indicated as “<0.001” instead of displaying “0.000.” This is a standard practice to avoid the implication of perfect precision.

Table 1: Typically, in Table 1, covariates are compared by exposure level. I suggest that the authors restructure this table to examine how the covariates differ by the exposure variable, which in this case would be serum ferritin levels.

Confounding Variables: The authors have only adjusted for age and pre-pregnancy BMI in the multivariable regression model. However, there are several other potential confounders that should be considered, such as blood pressure, physical activity levels, previous GDM diagnosis, sedentary lifestyle, and history of poor pregnancy outcomes. Could the authors explain why only age and BMI were included and how they plan to address the possible residual confounding from these unaccounted variables? More comprehensive adjustments would strengthen the robustness of the findings.

Discussion of Findings: The results indicate a positive association between serum ferritin (SF) and 1-hour plasma glucose (PG1H) levels, but a negative association between SF and HbA1c levels. The discussion around this unexpected finding is currently insufficient. I recommend that the authors explore and discuss more potential mechanisms or biological explanations that might account for this inverse relationship. This would help the readers better understand the broader implications of the findings.

Data Representation: It would also be beneficial to provide more clarity on how the statistical models were built and any assumptions tested (e.g., multicollinearity, linearity) to ensure the validity of the regression models.

Experimental design

no comment

Validity of the findings

Need to test for the assumption.

---

## Round 0.2 · accepted · Accept

· Academic Editor

Accept

The authors have revised the manuscript according to reviewers' suggestions.

·

Basic reporting

Literature references, sufficient background is provided.

Experimental design

Research question well defined, relevant & meaningful. It is stated how research fills an identified knowledge gap.
Methods described with sufficient detail & information to replicate..

Validity of the findings

Conclusions are well stated, linked to original research question & limited to supporting results.

Additional comments

None.